# Concerted EP2 and EP4 Receptor Signaling Stimulates Autocrine Prostaglandin E_2_ Activation in Human Podocytes

**DOI:** 10.3390/cells9051256

**Published:** 2020-05-19

**Authors:** Eva Mangelsen, Michael Rothe, Angela Schulz, Aikaterini Kourpa, Daniela Panáková, Reinhold Kreutz, Juliane Bolbrinker

**Affiliations:** 1Charité – Universitätsmedizin Berlin, Corporate Member of Freie Universität Berlin, Humboldt-Universität zu Berlin, and Berlin Institute of Health, Institute of Clinical Pharmacology and Toxicology, Charitéplatz 1, 10117 Berlin, Germany; eva.mangelsen@charite.de (E.M.); angela-martina.schulz@charite.de (A.S.); reinhold.kreutz@charite.de (R.K.); 2Lipidomix GmbH, Robert-Rössle-Str. 10, B55, 13125 Berlin, Germany; michael.rothe@lipidomix.de; 3Max Delbrück Center for Molecular Medicine in the Helmholtz Association, Electrochemical Signaling in Development and Disease, Robert-Rössle-Str. 10, 13125 Berlin, Germany; Aikaterini.Kourpa@mdc-berlin.de (A.K.); Daniela.Panakova@mdc-berlin.de (D.P.); 4DZHK (German Centre for Cardiovascular Research), Partner Site Berlin, Potsdamer Straße 58, 10785 Berlin, Germany

**Keywords:** podocyte, hyperfiltration, chronic kidney disease, prostaglandin E2, COX2, EP2, EP4, G protein-coupled receptor (GPCR) signaling, LC/ESI-MS/MS, MWF, SHR

## Abstract

Glomerular hyperfiltration is an important mechanism in the development of albuminuria. During hyperfiltration, podocytes are exposed to increased fluid flow shear stress (FFSS) in Bowman’s space. Elevated Prostaglandin E2 (PGE_2_) synthesis and upregulated cyclooxygenase 2 (Cox2) are associated with podocyte injury by FFSS. We aimed to elucidate a PGE_2_ autocrine/paracrine pathway in human podocytes (hPC). We developed a modified liquid chromatography tandem mass spectrometry (LC/ESI-MS/MS) protocol to quantify cellular PGE_2_, 15-keto-PGE_2_, and 13,14-dihydro-15-keto-PGE_2_ levels. hPC were treated with PGE_2_ with or without separate or combined blockade of prostaglandin E receptors (EP), EP2, and EP4. Furthermore, the effect of FFSS on *COX2*, *PTGER2*, and *PTGER4* expression in hPC was quantified. In hPC, stimulation with PGE_2_ led to an EP2- and EP4-dependent increase in cyclic adenosine monophosphate (cAMP) and *COX2*, and induced cellular PGE_2_. *PTGER4* was downregulated after PGE_2_ stimulation in hPC. In the corresponding LC/ESI-MS/MS in vivo analysis at the tissue level, increased PGE_2_ and 15-keto-PGE_2_ levels were observed in isolated glomeruli obtained from a well-established rat model with glomerular hyperfiltration, the Munich Wistar Frömter rat. *COX2* and *PTGER2* were upregulated by FFSS. Our data thus support an autocrine/paracrine COX2/PGE_2_ pathway in hPC linked to concerted EP2 and EP4 signaling.

## 1. Introduction

Podocytes are terminally differentiated epithelial cells that form the third layer of the glomerular filter with their interdigitating foot processes [1]. Their high degree of differentiation permits podocytes to accomplish their highly specialized functions. However, it limits their regenerative capacity, making them particularly vulnerable to pathological conditions such as glomerular hyperfiltration. When nephron number is reduced, compensatory changes of the remaining functional nephrons lead to adaptation of glomerular hemodynamics, resulting in increased glomerular filtration rate (GFR) in the single nephron and concomitantly in higher ultrafiltrate flow in Bowman’s space [2,3,4] (reviewed in [5]). This causes increased fluid flow shear stress (FFSS) and contributes to podocyte damage [3,6]. Perturbation of the glomerular filtration barrier contributes to proteinuria, glomerulosclerosis, and alteration in GFR, and thus promotes the gradual decline in renal function as observed in chronic kidney disease (CKD) (reviewed in [7,8,9]).

Understanding the pathomechanisms underlying podocyte damage due to glomerular hyperfiltration might help to identify therapeutical targets to protect against maladaptive responses of podocytes, which otherwise contribute to renal damage. Previous studies support a pathophysiological role of Cox2 (Ptgs2, cyclooxygenase 2) and prostaglandin E2 (PGE_2_) activation for development of albuminuria by increasing the permeability of the glomerular filtration barrier (reviewed in [5]). Furthermore, upregulation of Cox2 and Ptger2 (prostaglandin E receptor 2, EP2) was shown in uninephrectomized mice and murine podocytes exposed to FFSS, i.e., in two different experimental settings to study hyperfiltration [10]. These data suggest that PGE_2_ synthesis and signaling may play a role in podocyte responses to hyperfiltration. 

Cox2 is long known to mediate increased synthesis of PGE_2_ upon diverse stimuli (reviewed in [11,12]). Extracellularly, PGE_2_ exerts its effects via four different G-protein coupled prostaglandin E receptors (EP1–4) in human and rodents (reviewed in [13]). EP1, -2 and -4 mRNA expression was reported in mouse podocytes, EP2 and -4 were also detected on protein level [14]. However, the expression of EP in human podocytes (hPC) is unclear. Both EP2 and EP4 stimulate adenylate cyclase activity leading to elevated cyclic adenosine monophosphate (cAMP) levels while EP1 increases intracellular Ca^2+^ (reviewed in [15,16,17]). 

An autocrine/paracrine pathway between PGE_2_ and Cox2 was described previously, indicating that PGE_2_ leads to upregulation of Cox2 in osteocyte-like cells (murine long bone osteocyte Y4, MLO-Y4) [18]. This, in turn, increases synthesis of intracellular PGE_2_, which again induces Cox2. It remains a matter of debate which EP mediates this mutual amplification: In mouse podocytes, EP4 and the p38 mitogen-activated protein kinase (MAPK) signaling pathway were described to be involved in PGE_2_-mediated Cox2 upregulation and cAMP increase [19]. In other cell types, activation of EP2 with or without EP4-coupled cAMP/proteinkinase A (PKA) pathway was shown to upregulate Cox2 following PGE_2_-treatment [20,21,22]. EP2 signaling was shown to be the relevant mechanism in response to FFSS in mouse podocytes [14]. Indeed, the PGE_2_-Cox2-EP2 axis is suggested to be the relevant target for podocyte damage induced by FFSS [10,23]. So far, the mechanisms involved have not been investigated in hPC in detail. In this study, we therefore aimed to elucidate an autocrine/paracrine PGE_2_/COX2 pathway in hPC and to identify which EP contributes to this crosstalk. We determined *COX2, PTGER2,* and *PTGER4* expression in hPC after PGE_2_ stimulation and FFSS. Our results corroborate recent findings in murine models of hyperfiltration on autocrine/paracrine Cox2 and PGE_2_ activation in hPC. Moreover, we find this pathway in hPC to be linked to concerted EP2 and EP4 signaling. 

Importantly, distinct analysis of cellular PGE_2_ and its metabolites is crucial to elucidate their pathophysiological role in podocyte damage [10,23]. However, precise measurement of intracellular prostaglandins remains challenging. Enzyme-linked immunosorbent assays (ELISA) are widely used but have their limitations, e.g., the lack of standardization across different kits and low specificity, selectivity, and throughput compared to liquid chromatography tandem mass spectrometry (LC-MS/MS) methods [24,25]. As a limitation, LC-MS/MS oftentimes requires large quantities of samples which are difficult to obtain in cell culture experiments [26,27,28,29,30,31,32]. We were able to overcome these obstacles and provide an approach to analyze prostaglandins in hPC by liquid chromatography electrospray ionization tandem mass spectrometry (LC/ESI-MS/MS). With our modified LC/ESI-MS/MS protocol, we were able to precisely quantify cellular PGE_2_, 15-keto-PGE_2_, and 13,14-dihydro-15-keto-PGE_2_ levels. After stimulation with PGE_2_, the cellular PGE_2_-content was elevated, which was completely blocked by pharmacological inhibition EP2 and EP4. In addition, we performed corresponding in vivo analysis at the tissue level by using the LC/ESI-MS/MS methodology and demonstrated increased PGE_2_ and 15-keto-PGE_2_ levels in isolated glomeruli obtained from a well-established rat model with glomerular hyperfiltration, i.e., the Munich Wistar Frömter rat (MWF).

Our findings on elevated glomerular PGE_2_ and 15-keto-PGE_2_ levels strengthen the hypothesis that glomerular PGE_2_-induction associates with albuminuria due to podocyte damage.

## 2. Materials and Methods

### 2.1. Cell Culture

Conditionally immortalized hPC (kindly provided by Moin A. Saleem, University of Bristol, UK) were cultured according to the original protocol [33,34] with slight modifications. The cells proliferate at 33 °C and transform to differentiated hPC when kept at ≥37 °C exhibiting podocyte-specific markers [34]. Briefly, podocytes were grown at 33 °C and 5% CO_2_ in Roswell Park Memorial Institute (RPMI)-1640 medium (cat. no. BS.F1215, Bio&SELL, Feucht/Nürnberg, Germany) supplemented with 1% Insulin-Transferrin-Selenium 100X (cat. no. 41400-045, Gibco, Grand Island, NY, USA), 10% fetal bovine serum (FBS, cat. no. F7524, Sigma, Steinheim, Germany) and 1% ZellShield^®^ to prevent contamination (cat. no. 13-0150, Minerva Biolabs, Berlin, Germany). Medium was changed 2–3 times per week. At confluency of 70–80%, podocytes were transferred to 37–38 °C until full confluence and proliferation arrest. Subsequently, cells were kept for a minimum of 14 days at 37–38 °C to obtain full differentiation. Differentiated phenotype was confirmed by analysis of the marker synaptopodin by immunofluorescence (see Appendix A). Characterization also included overall comparison of the cellular shape (“cobblestone-like” in undifferentiated state and “arborized” in differentiated hPC [33]) by light microscopy, synaptopodin mRNA expression, as well as nephrin and podocin protein detection by immunofluorescence and western blot (see Appendix A). Prior to experiments, cells were detached with Trypsin 0.25%/EDTA 0.02% solution (cat. no. L-2163, Biochrom, Berlin, Germany), seeded in 12-well plates at 1 × 10^5^ cells per well and kept in RPMI-1640 medium with supplements for adherence overnight. All experimental treatments were carried out in supplement-free RPMI-1640 medium at 37–38 °C with cell passages between 5 and 22.

### 2.2. PGE_2_ Treatment and Inhibition of EP Receptors

PF-04418948 (cat. no. PZ0213, Sigma, Steinheim, Germany) served as EP2 antagonist [35,36] and ONO-AE3-208 (cat. no. 14522, Cayman Chemical, Ann Arbor, MI, USA) was chosen as EP4 antagonist [37,38]. Stock solutions of PGE_2_ (cat. no. 14010, Cayman Chemical, Ann Arbor, MI, USA), PF-04418948, and ONO-AE3-208 with 10 mM were prepared in DMSO (cat. no. D2650, Sigma, Steinheim, Germany) and stored at −20 °C until further use. 

Podocytes were treated with PGE_2_ at 10 nM–1 µM concentrations as PGE_2_-concentrations up to 1 µM are commonly used for in vitro experiments in murine podocytes [39,40]. For inhibition experiments, 1 µM or even higher concentrations of the selective EP2 and/or EP4 antagonist were used in previous studies [36,37,41,42,43]. In a pilot study, treatment with PGE_2_ and EP2 antagonist (1 µM each) did not show inhibitory effects (Appendix A). Thus, antagonists were added concomitantly to PGE_2_ 100 nM for the indicated time-points. 

### 2.3. Determination of Intracellular cAMP Levels

Intracellular cAMP levels were measured using an ELISA kit (cat. no. ADI-901-163, Enzo Life Sciences, Farmingdale, NY, USA). Cells were lysed in 300–400 µL 0.1 M HCl containing 0.1% Triton X-100 and samples were processed according to the manufacturer’s instructions for the non-acetylated format. PGE_2_ stimulated samples were diluted 1:5, and samples of PGE_2_ stimulation plus co-incubation with either the EP2 or the EP4 antagonist were diluted 1:2–3 in lysis buffer. Optical density was measured at 415 nm and cAMP concentrations were normalized for protein content for each sample. Protein amount was quantified by a colorimetric kit (cat. no. 23227, Pierce^TM^ BCA Protein Assay Kit, Thermo Fisher Scientific, Rockford, IL, USA). Experiments for cAMP consisted of *n* = 3–6 samples per experimental group and were performed once or in duplicate as indicated.

### 2.4. Reverse Transcription and Quantitative Real-Time PCR

Total RNA of hPC was isolated using the RNeasy^®^ Micro Kit (cat. no. 74004, Qiagen, Hilden, Germany) following the manufacturer’s protocol. RNA quality was controlled by a 260/280 nm absorption ratio. For cDNA synthesis, total RNA was reverse-transcribed using the First Strand cDNA Synthesis Kit (cat. no. K1612, Thermo Fisher Scientific, Vilnius, Lithuania). 

Quantitative Real-Time PCR (qPCR) was conducted in a CFX96 Touch PCR system (Bio-Rad, München, Germany; software version 3.1.1517.0823) or in a 7500 Fast Real-Time PCR System (Applied Biosystems, Darmstadt, Germany; software version 2.0.6) using the comparative quantitative cycle method with SYBR-green (cat. no. 4,385,612 and 100029284, Thermo Fisher Scientific, Vilnius, Lithuania) as reported previously [44,45] Expression analysis of each sample was done in three technical replicates and only samples with an intra-triplicate standard deviation (SD) < 0.2 were used for further calculation. Normalization of expression was done by the reference gene glyceraldehyde 3-phosphate dehydrogenase *(GAPDH)*. ΔΔCt was normalized to the untreated controls in hPC. All results were plotted as log2 of fold change (FC) (2^-ΔΔCt). Primer sequences are listed in Table 1. Primers were purchased from Eurofins Genomics, Ebersberg, Germany or Tib Molbiol, Berlin, Germany and specificity of detected reverse transcriptase (RT)-PCR products was confirmed by sequencing at Eurofins Genomics, Ebersberg, Germany. *COX2*-qPCR for hPC consisted of *n* = 3–8 samples per experimental group and were performed in duplicate or triplicate as indicated. 

### 2.5. LC/ESI-MS/MS for Analysis of Prostaglandins

#### 2.5.1. Sample Preparation 

After stimulation or inhibition experiments, supernatants were removed and stored at −80 °C. Cells were washed twice with cold phosphate buffered saline (PBS), PBS was completely aspirated from wells, and the 12-well plates were immediately stored at −80 °C until further processing. Before analysis, cells were scraped from plate and suspended in 500 µL water. A 50 µL aliquot was taken for total protein measurement following the Lowry protocol.

The cell suspensions were spiked with an internal standard consisting of 14,15-Epoxyeicosatrienoic acid-d8, 14,15-Dihydroxyeicosatrienoic acid-d11, 15-Hydroxyeicosatetraenoic acid-d8, 20-Hydroxyeicosatetraenoic acid-d6, Leukotriene B_4_-d4, PGE_2_-d4 1 ng each (Cayman Chemical, Ann Arbor, MI, USA). In addition, 500 µL methanol and 5 µL 2,6-di-tert-butyl-4-methylphenol (BHT, 10 mg/mL) were added and shaken vigorously.

The total prostaglandins were released using phospholipase A2 from honey bee Apis Mellifera (Sigma-Aldrich, Taufkirchen, Germany) as described previously [46]. After pH adjustment to 6, acetic acid samples were extracted by solid phase extraction (SPE) using Bond Elute Certify II columns (Agilent Technologies, Santa Clara, CA, USA), which were preconditioned with 3 mL methanol, followed by 3 mL of 0.1 mol/L phosphate buffer containing 5% methanol (pH 6). SPE-columns were then washed with 3 mL methanol/H_2_O (40/50, *v*/*v*). For elution, 2 mL of n-hexane:ethyl acetate 25:75 with 1% acetic acid was used. The extraction was performed with an SPE Vacuum Manifold. The eluate was evaporated on a heating block at 40 °C under a stream of nitrogen to obtain a solid residue which was dissolved in 100 µL methanol/water 60:40 and transferred in an HPLC autosampler vial (HPLC, high performance liquid chromatography).

Experiments for analysis of prostaglandins in hPC consisted of *n* = 3–6 replicates per experimental group and. Experiments were performed once or in triplicate (on different cell passages and on different days) as indicated.

Rat glomeruli obtained by differential sieving of one kidney as described below were divided into 3 parts and stored at −80 °C until further analysis. One aliquot with approximately 1/3 total kidney was prepared as described for cells, but without application of phospholipase A2. 

For rat plasma, 200 µL plasma were spiked with internal standard and BHT. In addition, 20 µL glycerol and 500 µL acetonitrile was added and shaken vigorously. pH was adjusted at 6 with 2 mL phosphate buffer (0.1 mol/L). The samples were centrifuged and the clear supernatant was extracted using SPE as described above.

Experiments for analysis of prostaglandins in rat glomeruli or plasma consisted of *n* = 8–10 glomerular isolated or plasma samples of *n* = 8–10 different animals per rat strain. 

#### 2.5.2. LC/ESI-MS/MS

The residues were analyzed using an Agilent 1290 HPLC system with binary pump, multisampler and column thermostat with a Zorbax Eclipse plus C-18, 2.1 × 150 mm, 1.8 µm column using a solvent system of aqueous acetic acid (0.05%) and acetonitrile. The elution gradient was started with 5% organic phase, which was increased within 0.5 min to 32%, 16 min to 36.5%, 20 min to 38%, 28 min to 98% and held there for 5 min. The flow rate was set at 0.3 mL/min, the injection volume was 20 µL. The HPLC was coupled with an Agilent 6495 Triplequad mass spectrometer (Agilent Technologies, Santa Clara, CA, USA) with electrospray ionisation source. The source parameters were Drying gas: 115 °C/16 L/min, Sheath gas: 390 °C/12 L/min, Capillary voltage: 4300 V, Nozzle voltage: 1950 V, and Nebulizer pressure: 35 psi.

Analysis was performed with Multiple Reaction Monitoring in negative mode. For details, see Appendix A. Unless stated otherwise, all solvents and chemicals were purchased from VWR International GmbH, Darmstadt, Germany.

### 2.6. FFSS

For FFSS, 1 × 10^5^–6 × 10^5^ cells where seeded on collagen IV coated Culture Slips^®^ (cat. no. CS-C/IV, Dunn Labortechnik GmbH, Asbach, Germany), which are glass slides coated with collagen type IV and rimmed with a 1.0 mm wide polytetrafluoroethylene border to limit cell culture growth to the portion of the slip exposed to fluid flow. FFSS experiments were performed as previously described with slight modifications [23]. The Streamer^®^ Shear Stress Device (cat. no. STR-400, Dunn Labortechnik GmbH, Asbach, Germany) was installed in a 38 °C incubator with 5% CO_2_ and prepared as follows: 400 mL of PBS followed by RPMI-1640 medium were pumped through the device for approximately 10 min each. Prior to each change of content, flow direction was reversed to empty the tubes from the previous liquid. After washing, medium was replaced by 400 mL of new medium. The system was checked for leaks and air bubbles were eliminated. After preparation of the streamer, flow direction was again reversed until the streamer was half-filled by medium. Tubes were released from the pump, the system was taken out of the incubator, Culture Slips^®^ with hPC were inserted in the Streamer^®^, and the system was placed back into the incubator. All 6 slots of the Streamer^®^ were filled to allow consistent flow. Based on previous research, we applied FFSS at 2 dynes/cm^2^ for 2 h [23]. At the end of each experiment, flow rate was reversed and cells were released from the device. Control cells were put in the same incubator with the same medium but were not exposed to FFSS. 

### 2.7. Animals

The MWF rat served as a model for CKD with albuminuria, while the spontaneously hypertensive rat (SHR) served as a control strain. SHR rats develop hypertension early in life but are resistant to albuminuria development as reviewed in [47]. 

Male rats at 8 weeks of age were deployed from our MWF/Rkb (RRID:RGD_724569, laboratory code Rkb https://www.nationalacademies.org/ilar/lab-code-database) and SHR/Rkb (RRID:RGD_631696, laboratory code Rkb https://www.nationalacademies.org/ilar/lab-code-database) colonies at Charité—Universitätsmedizin Berlin, Germany. Rats were kept under standard conditions as described previously [44]. All experimental work in rat models was performed in accordance with the guidelines of the Charité—Universitätsmedizin Berlin and the local authority for animal protection (Landesamt für Gesundheit und Soziales, Berlin, Germany) for the use of laboratory animals. The registration numbers for the rat experiments are G 130/16 (approved 2 August 2016) and T 0189/02 (approved 31 August 2018). Anesthesia was achieved by ketamine-xylazine (87 and 13 mg/kg body weight, respectively). Kidneys were obtained, decapsulated, and sieved using a 125 µm steel sieve (Retsch GmbH, Haan, Germany) rinsed by PBS. The filtrate was then placed on a 71 µm steel sieve (Retsch GmbH, Haan, Germany) and washed with PBS. Glomeruli were kept on the sieve and were separated from the flow-through. Glomeruli were rinsed off the sieve with PBS, centrifuged, snap-frozen, and stored at −80 °C until further processing. Plasma was obtained by retrobulbary punction or punction of vena cava and collected in ethylenediaminetetraacetic acid (EDTA)-containing vials, centrifuged at 2 min at 4 °C, and stored subsequently at −80 °C. 

### 2.8. Statistics

Statistical analysis was conducted using GraphPad Prism 8.4.0 (GraphPad Software, San Diego, CA, USA). Normal distribution was tested with the Shapiro–Wilk test. Normally distributed data were compared either by unpaired, two-tailed Student’s *t*-test or one-way ANOVA with Tukey’s or Dunnett’s multiple comparisons test as indicated. Multiple comparisons tests after one-way ANOVA were used to compare every mean to every other mean (Tukey’ follow up test) or to a control mean (Dunnett’s follow up test). Results not normally distributed were analyzed by Mann–Whitney test or Kruskal–Wallis test with Dunn’s multiple comparisons test as indicated. Significance level was set at *p* < 0.05. Statistical details for specific experiment can be found within figures and figure legends.

## 3. Results

### 3.1. PGE_2_ Leads to EP2- and EP4- Dependent Increased cAMP Levels in Differentiated hPC 

Stimulation of hPC with 100 nM PGE_2_ led to an immediate time-dependent increase in intracellular cAMP levels detected after 1 min onward and retained at least until 40 min of PGE_2_ stimulation (Figure 1). As intracellular cAMP levels remained comparably high until 20 min of PGE_2_ stimulation, this incubation time was chosen for subsequent cAMP measurements.

Analysis of EP expression on hPC revealed the presence of *PTGER1, PTGER2*, and *PTGER4* mRNA in differentiated hPC (Appendix A), which encode for EP1, EP2, and EP4, respectively. As only EP2 and EP4 are reported to mediate an increase in intracellular cAMP (reviewed in [15,16,17]), we next investigated the effect of pharmacological inhibition of EP2 and EP4 signaling on PGE_2_-stimulated intracellular cAMP levels in hPC. Therefore, either the selective antagonist of EP2 (PF-04418948, 1 µM) or EP4 (ONO-AE3-208, 1 µM) were co-incubated with 100 nM PGE_2_ individually and in combination (Figure 2). Upon PGE_2_ stimulation, antagonism of either EP2 (−92.5%) or EP4 (−63.7%) alone resulted in a marked albeit only partial decrease of intracellular cAMP levels compared to stimulated hPC without antagonists. In contrast, the PGE_2_ stimulated intracellular cAMP increase was completely abrogated by combined EP2 and EP4 antagonism (Figure 2), suggesting that, in hPC, both EP2 and EP4 may mediate PGE_2_-dependent signaling.

### 3.2. PGE_2_ Induces COX2 Gene Expression via EP2 and EP4 Signaling in Differentiated hPC

Stimulation of hPC with PGE_2_ for 2 h revealed a dose-dependent upregulation of *COX2* mRNA expression (Figure 3a). In order to elucidate the role of EP2 and EP4 in PGE_2_-mediated *COX2* upregulation, either the selective EP2 antagonist PF-04418948 (1 µM) or the selective EP4 antagonist ONO-AE3-208 (1 µM) were co-incubated with 100 nM PGE_2_ individually or in combination (Figure 3b). Upon PGE_2_ stimulation, antagonism of either EP2 or EP4 alone resulted in an increase in *COX2* mRNA although lower compared to PGE_2_-stimulated hPC without antagonists (Figure 3b). Combined EP2 and EP4 antagonism completely inhibited PGE_2_-mediated *COX2* upregulation (Figure 3b), suggesting that PGE_2_ signals via both EP2 and EP4 to regulate *COX2* levels in a positive feedback loop. 

### 3.3. PGE_2_ Reduces PTGER2 and PTGER4 Gene Expression Which Is Not Modified by EP2 or EP4 Antagonists in Differentiated hPC

Stimulation of hPC with rising concentrations of PGE_2_ for 2 h revealed inconsistent changes of *PTGER2* mRNA expression: 10 nM and 1 µM did not significantly change *PTGER2* expression, whereas PGE_2_ 100 nM slightly reduced *PTGER2* expression (Figure 4a). The weak downregulation by *PTGER2* of 100 nM PGE_2_ was not abrogated by co-incubation with the selective EP4 antagonist ONO-AE3-208 (1 µM) (Figure 4b). 

Stimulation of hPC with PGE_2_ for 2 h revealed a dose-dependent reduction of *PTGER4* mRNA expression (Figure 4c). In order to elucidate the role of EP2 in PGE_2_-mediated *PTGER4* downregulation, the selective EP2 antagonist PF-04418948 (1 µM) was co-incubated with 100 nM PGE_2_ (Figure 4d). The PGE_2_-induced decrease in *PTGER4* mRNA was not abolished by co-incubation with the EP2 antagonist (Figure 4d).

### 3.4. Cellular PGE_2_ and Metabolite Profile in hPC after PGE_2_ Stimulation: Effects of EP2 and EP4 Blockade

To investigate whether PGE_2_ stimulation and subsequent *COX2* induction lead to changes in cellular levels of PGE_2_ and its downstream metabolites 15-keto-PGE_2_ and 13,14-dihydro-15-keto-PGE_2_ (Figure 5a), hPC were analyzed by LC/ESI-MS/MS. After stimulation with PGE_2_, the cellular PGE_2_-content was elevated (Figure 5b), while 15-keto-PGE_2_ and 13,14-dihydro-15-keto-PGE_2_ remained at control levels. Pharmacological inhibition of EP2 and EP4 reduced cellular PGE_2_ significantly (Figure 5c). Our findings point towards an autocrine PGE_2_-EP2/EP4-COX2 signaling axis in hPC.

### 3.5. Glomerular PGE_2_ and Metabolite Profile in Glomeruli and Plasma in the CKD MWF Model

Analysis of PGE_2_ and its subsequent metabolites 15-keto-PGE_2_ and 13,14-dihydro-15-keto-PGE_2_ in glomeruli and plasma of MWF and SHR at eight weeks of age revealed an increase of glomerular PGE_2_ and 15-keto-PGE_2_ levels in MWF compared to SHR. No difference was observed for 13,14-dihydro-15-keto-PGE_2_ (Figure 6a). In plasma, levels of PGE_2_ and 13,14-dihydro-15-keto-PGE_2_ did not differ between MWF and SHR (Figure 6b). The metabolite 15-keto-PGE_2_ in plasma was below level of detection (data not shown).

### 3.6. FFSS Increases COX2 and PTGER2 Gene Expression in hPC

FFSS was previously shown to elevate intracellular PGE_2_ levels and Cox2 in murine podocytes [10,23]. We therefore investigated *COX2* mRNA expression after FFSS in hPC. FFSS led to increased *COX2* mRNA expression in hPC as shown in Figure 7.

Furthermore, EP2 protein was reported to be upregulated upon FFSS in murine podocytes [10]. In hPC, FFSS slightly upregulated *PTGER2* (Figure 8a), whereas *PTGER4* expression did not change compared to control (Figure 8b).

## 4. Discussion

Recent studies in murine models of hyperfiltration support a pathophysiological role of autocrine/paracrine COX2/PGE_2_ activation on podocyte damage, thus contributing to disturbances of the glomerular filtration barrier including the development of albuminuria [10,14,23] (reviewed in [2,5]). These findings suggest that induction of COX2 associates with podocyte damage, while selective or non-selective inhibition of COX2 reduces proteinuria in animal models as well as in patients (reviewed in [48]). So far, the mechanisms involved have not been investigated in detail. 

In mouse podocytes, EP1, -2 and -4 expression was reported and EP2 and -4 were also detected on protein level [14]. In this study, we corroborate these findings in hPC, which also express EP1, -2 and -4. EP4 is known as a constitutively expressed protein reflected by abundant protein levels in untreated murine podocytes compared to EP2 [10]. Our results on apparently lower *PTGER4* mRNA expression compared to *PTGER2* in hPC should be interpreted with caution as they need to be confirmed on the protein level in future investigations. Stimulation of hPC with PGE_2_ led to an immediate intracellular cAMP increase starting at 1 min after PGE_2_ stimulation until at least 40 min of stimulation (Figure 1). Of note, the detected intracellular cAMP levels are net levels resulting from cAMP generation by adenylate cyclase and its concomitant degradation by phosphodiesterases. Phosphodiesterase activity was only blocked at the end of the stimulation experiments by adding HCl. Previous studies in immortalized murine podocytes revealed a similar time-course of cAMP increase occurring within the first 30 min after EP2 and/or EP4 stimulation [39,49,50]. In our experimental setting in hPC, this PGE_2_-stimulated intracellular cAMP increase was only completely abrogated by combined EP2 and EP4 antagonism pointing towards a comparable role of both receptors for cAMP induction in hPC (Figure 2). Multiple intracellular signaling pathways have been described for either EP2 and EP4 (reviewed in [51]). PGE_2_ stimulation of EP2 and EP4 activated the transcription factors T-cell factor (Tcf) and lymphocyte enhancer factor (Lef) signaling via PKA- and phosphatidylinositol 3-kinase/proteinkinase B (PI3K/Akt)-dependent phosphorylation of glycogen synthase kinase 3 (GSK3), thus promoting translocation of the transcription cofactor β-catenin into the nucleus where interaction with Tcf and Lef modulated gene expression, e.g., of COX2 [52,53]. However, participation of EP2 in PI3K/Akt signaling remains a matter of debate, as some investigators suggest that only EP4 but not EP2 are linked with PI3K/Akt [54]. Mechanotransduction in murine podocytes was previously suggested to be mediated by Akt-GSK3β-β-catenin, extracellular-signal regulated kinases (ERK)1/2, and p38MAPK, but not cAMP-PKA signaling upon FFSS [40]. The lack of cAMP elevation upon FFSS in that study might be explained, though by the experimental design as intracellular cAMP was measured at the earliest 2 h after applying FFSS. In contrast, the cAMP-PKA pathway was shown to be involved upon PGE_2_ stimulation of murine podocytes [40], which better matches our setting of PGE_2_ stimulation of hPC. Moreover, the cAMP-PKA pathway has been shown in intracellular signaling upon FFSS in osteocytes [55,56]. 

We detected upregulation of *COX2* by PGE_2_ in a dose-dependent fashion with both EP2 and EP4 being involved in hPC (Figure 3). PGE_2_ stimulation of EP2 and EP4 was reported to increase cAMP response element-binding protein (CREB), which was demonstrated to be PKA-dependent for EP2, whereas EP4-coupled PI3K signaling was suggested to counteract CREB formation [57,58,59,60]. Of note, transcription of *COX2* can be modulated by CREB, as CRE is part of the COX2 promotor [61,62]. Therefore, subsequent PKA/CREB activation could play a role for the observed increase in *COX2* expression following intracellular cAMP level elevation in hPC. To further investigate this aspect, experiments with PKA-inhibitors, e.g., H-89, will be performed as well as analysis of CREB phosphorylation status, and activation of the transcription factors Tcf and Lef. Functional analysis of COX2 protein activity might also be helpful. However, previous data in various cell types including murine podocytes already revealed that COX2 protein is indeed increased after 2 h stimulation with PGE_2_ [19,58]. Taken together, our results on concerted EP2 and EP4 signaling being involved in upregulation of intracellular cAMP and *COX2* levels represent a novel finding. To validate our results on the role of EP2 and EP4 on the intracellular cAMP increase and upregulation of *COX2*, effects of hPC stimulation with an EP2 and/or EP4 agonist without PGE_2_ should be investigated in future experiments.

Accompanied by these findings, PGE_2_ stimulation also increased cellular PGE_2_, i.e., PGE_2_, which is released from membranes and appears intracellularly (Figure 5b). This effect is abrogated by combined EP2- and EP4-antagonism (Figure 5c). Intracellularly generated PGE_2_ is degraded by 15-prostaglandin dehydrogenase (HPGD) to 15-keto-PGE_2_, which is then terminally inactivated, albeit with different efficiency, by prostaglandin reductase (PTGR) 1, -2 and -3 to 13,14-dihydro-15-keto-PGE_2_ [63,64] (reviewed in [65]). Intracellular PGE_2_ was reported to exit the cell by simple diffusion or by an efflux transport mediated by prostaglandin transporter (PGT), i.e., solute carrier organic anion transporter family, member 2A1 (OATP2A1), or ATP-binding cassette, subfamily C, member 4 (MRP4) [66,67,68].

Elevated PGE_2_ levels were previously associated with podocyte damage, suggesting that it might be a biomarker of progressive CKD [10,69]. LC-MS/MS based methods are beneficial to precisely study cellular prostaglandin metabolism with a maximum of selectivity and specificity. However, cell culture experiments are mainly restricted to small sample amounts that might hamper analysis by LC-MS/MS [26,27,28,29,30,31,32]. Here, we present a refined protocol for prostaglandin analysis in cells by LC/ESI/MS-MS. This might help to further elucidate cellular prostaglandin metabolism under pathophysiological conditions particularly in vitro but also in vivo. The observed increases in cellular PGE_2_ content upon PGE_2_ stimulation in hPC might be due to several mechanisms. One possibility is that extracellular PGE_2_ enters the podocyte by simple diffusion or by uptake transport mediated by OATP2A1, which was previously reported to facilitate bidirectional transport of PGE_2_ over membranes [66,67,68]. A second reason could be autocrine/paracrine mechanisms, i.e., extracellular PGE_2_ activates EP2 and EP4 signaling, thus increasing COX2 transcription and translation. As COX2 delineates the rate-limiting step of PGE_2_ synthesis (reviewed in [70]), its induction leads to higher cellular PGE_2_ levels. Our results support the latter option, as combined inhibition of EP2 and EP4 signaling diminished the increase in cellular PGE_2_ levels. 

In our corresponding in vivo study, we employed the MWF model, which represents a suitable model with glomerular hyperfiltration and thus FFSS [47]. The MWF model was previously extensively characterized (reviewed [47]). Thus, the MWF model is a non-diabetic inbred, genetic model with an inherited nephron deficit of 30–50% depending on the comparator rat strain [71,72]. Consequently, male MWF rats are characterized by increased single nephron glomerular filtration rate but with normal mean glomerular capillary pressure [72,73]. In addition, MWF rats develop mild arterial hypertension and spontaneous progressive albuminuria [47]. Thus, male MWF rats develop spontaneous albuminuria at an early age between weeks 4 and 8 after birth and subsequently progressive proteinuria and glomerulosclerosis [74]. The latter was also demonstrated in the MWF strain from our own colony and thus in the animals used in the current study [47]. Early onset albuminuria in young MWF animals occurs at six weeks of age and is preceded by glomerular hypertrophy, accompanied by focal and segmental loss of podoplanin and followed by podocyte foot process effacement at 8 weeks of age, i.e., at onset of albuminuria [75]. For these reasons, we selected animals at this age for our analysis in the current study. As a comparator strain, we use the previously characterized spontaneously hypertensive rats (SHR) that are resistant to albuminuria development [71,76]. Taken together, we showed the feasibility of the LC/ESI-MS/MS methodology to characterize the PGE_2_ pathway at the glomerular tissue level by using the MWF strain. The observed increases in both PGE_2_ and 15-keto-PGE_2_ in isolated glomeruli of MWF support the activation of this pathway in glomerular hyperfiltration. However, these results should be viewed against the background that the cellular origin of this finding was not determined, and thus the contribution of other cell types, e.g., glomerular endothelial cells or mesangial cells remains unclear. Up to now, direct isolation of podocytes from glomeruli was reported for transgenic mice [77,78,79,80]. In the rat, there seem to be more technical difficulties as transgenic implementation of fluorescent dyes was not yet accomplished. Antibody staining for podocyte markers and subsequent analysis by FACS is possible [81], but whether isolated primary rat podocytes can be subjected to transcriptomic, proteomic, or lipidomic analysis remains to be investigated. Urinary PGE_2_ was suggested as a biomarker for adaptive hyperfiltration in human solitary kidney [69]. The analysis of the urinary PGE_2_ and metabolite profile in our CKD MWF model is currently not established due to experimental challenges to establish robust LS/ESI-MS/MS analysis in rat urine. In contrast, the profile in plasma did not differ significantly between MWF and SHR (Figure 6b). In plasma, dilution of prostaglandins might be a major problem as 15-keto-PGE_2_ levels were below the limit of detection. Therefore, plasma levels provide rather a rough estimate, while analysis of glomeruli offers a closer insight into podocyte prostaglandin metabolism.

Besides mimicking hyperfiltration by exogenous supplementation with PGE_2_, we also applied FFSS on hPC. This model aimed to imitate intensified flow of the ultrafiltrate in Bowman’s space, thus causing podocyte injury. Similar to PGE_2_ treatment, FFSS leads to upregulation of *COX2* (Figure 7). Our data corroborate the work by Srivastava and coworkers, who suggested Akt/GSK3β-βcatenin and the MAPK pathway to be involved in mechanotransduction on mouse podocytes [10,40]. We thus aim to investigate these signaling pathways in hPC upon FFSS in our future work. We corroborate recent findings that EP2 is upregulated by FFSS while EP4 expression is not changed [10] (Figure 8). 

## 5. Conclusions

An autocrine/paracrine pathway between COX2 and PGE_2_ exists also in hPC and is mediated by both EP2 and EP4. Distinct analysis of cellular PGE_2_ and its metabolites was enabled by a modified protocol using LS/ESI-MS/MS. Elevated PGE_2_ and 15-keto-PGE_2_ levels were detected in glomeruli of MWF, a model for CKD, thereby strengthening the hypothesis that glomerular PGE_2_ accumulation is associated with albuminuria due to podocyte damage. Understanding prostaglandin signaling in hPC may contribute to identifying novel target pathways to protect against maladaptive responses to hyperfiltration in podocytes. 

## Figures and Tables

**Figure 1 cells-09-01256-f001:**
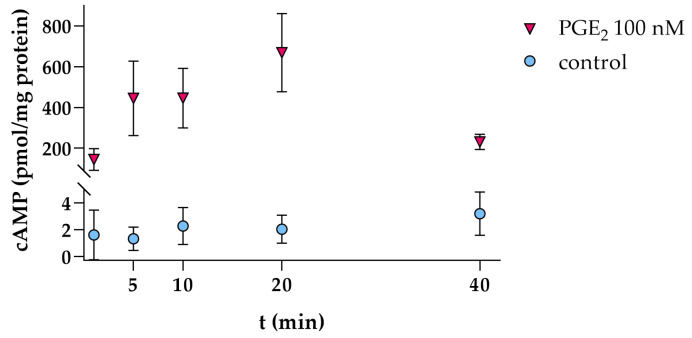
Intracellular cAMP levels in hPC are increased by PGE_2_ stimulation (100 nM, pink triangles) in a time-dependent manner. For several controls (blue circles), cAMP levels fell below the lowest concentration of the recommended standard curve (0.78 pmol/mL) and were therefore set to zero. Each data point represents the mean ± SD of one experiment with *n* = 3–4 samples per time-point.

**Figure 2 cells-09-01256-f002:**
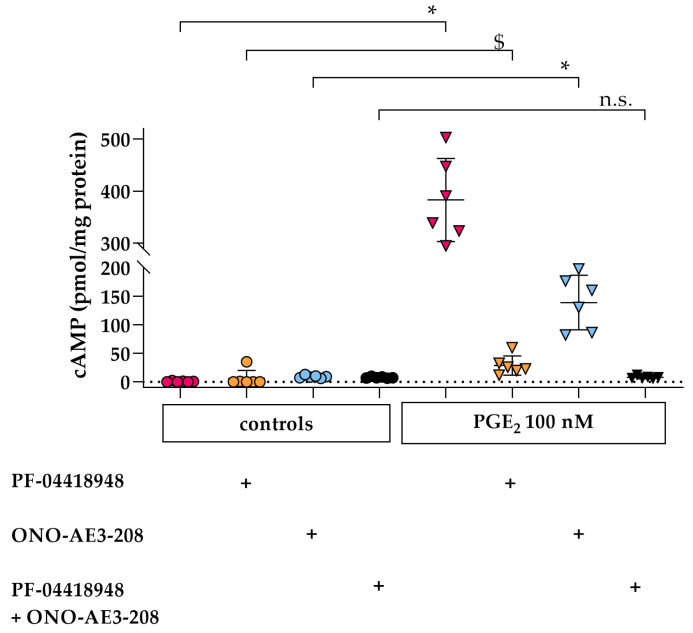
Intracellular cAMP is increased by PGE_2_ stimulation via EP2 and EP4 in hPC. Representative cAMP levels following PGE_2_ stimulation for 20 min without concomitant EP2 or EP4 antagonist (100 nM, pink triangles) compared to controls without PGE_2_ (pink circles), after co-incubation with either EP2 antagonist (PF-04418948, 1 µM, orange triangles) or EP4 antagonist (ONO-AE3-208, 1 µM, blue triangles) compared to controls without PGE_2_ (orange circles for EP2 antagonist, blue circles for EP4 antagonist), and co-incubation of PGE_2_ with both antagonists simultaneously (1 µM each, black triangles) compared to controls without PGE_2_ (black circles). Each data point represents a single sample and plotted as mean ± SD (horizontal lines) per treatment group consisting of *n* = 6 samples. For several controls, cAMP levels fell below the lowest concentration of the recommended standard curve (0.78 pmol/mL) and were therefore set to zero. Experiments were done in duplicate on different cell passages and on different dates, each consisting of *n* = 3–6 replicates per treatment, except for the separate EP4 inhibition, which was only performed once. Statistics: *, *p* < 0.01; $, *p* < 0.05; *n*.s., not significant, assessed by a Mann–Whitney test. + denotes addition of the respective EP antagonists.

**Figure 3 cells-09-01256-f003:**
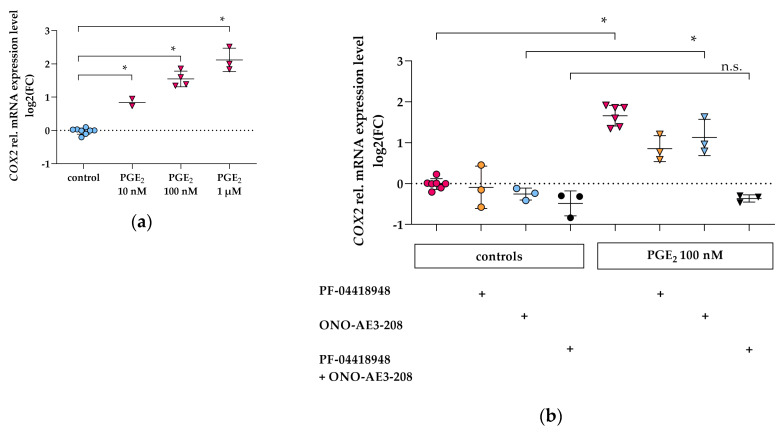
*COX2* gene expression is increased by PGE_2_ via EP2 and EP4 in hPC. qPCR results are presented as relative mRNA expression level normalized to *GAPDH* and referred to control group. (**a**) *COX2* levels following PGE_2_ stimulation (pink triangles) for 2 h were upregulated in a dose-dependent manner compared to untreated control (blue circles). Each data point represents the mean of an independent experiment (performed at least in duplicate on different cell passages and on different dates, each consisting of *n* = 3–8 replicates per treatment) and plotted as combined mean ± SD (horizontal lines). SD was not plotted when only two independent experiments were performed. Statistics: *, *p* < 0.01, assessed by one-way ANOVA with Dunnett’s follow-up test; (**b**) *COX2* levels following PGE_2_ stimulation for 2 h without concomitant EP2 or EP4 antagonist (100 nM, pink triangles) compared to controls without PGE_2_ (pink circles), after co-incubation with either EP2 antagonist (PF-04418948, 1 µM, orange triangles) or EP4 antagonist (ONO-AE3-208, 1 µM, blue triangles) compared to controls without PGE_2_ (orange circles for EP2 antagonist, blue circles for EP4 antagonist), and co-incubation of PGE_2_ with both antagonists simultaneously (1 µM each, black triangles) compared to controls without PGE_2_ (black circles) obtained in three independent experiments. Each data point represents the mean of an independent experiment (performed at least in triplicate on different cell passages and on different dates, each consisting of *n* = 3–6 replicates per treatment) and plotted as combined mean ± SD (horizontal lines). + indicates addition of the respective EP antagonists. Statistics: *, *p* < 0.01; *n*.s., not significant, assessed by two-tailed Student’s *t*-test.

**Figure 4 cells-09-01256-f004:**
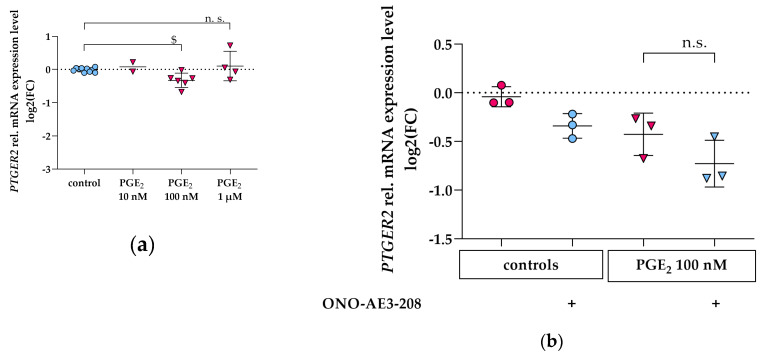
*PTGER2 and PTGER4* gene expression in hPC after PGE_2_ stimulation and following co-incubation with EP antagonists. qPCR results are presented as relative mRNA expression level normalized to *GAPDH* and referred to control group. (**a**) *PTGER2* levels following PGE_2_ stimulation with 10 nM, 100 nM, and 1 µM (pink triangles) for 2 h compared to untreated control (blue circles). Each data point represents the mean of an independent experiment (performed at least in duplicate on different cell passages and on different dates, each consisting of *n* = 3–8 replicates per treatment) and plotted as combined mean ± SD (horizontal lines). SD was not plotted when only two independent experiments were performed. Statistics: $, *p* < 0.05, assessed by one-way ANOVA with Dunnett’s follow-up test; (**b**) *PTGER2* levels following PGE_2_ stimulation for 2 h without concomitant EP4 antagonist (100 nM, pink triangles) compared to controls without PGE_2_ (pink circles), and after co-incubation with EP4 antagonist (ONO-AE3-208, 1 µM, blue triangles) compared to controls without PGE_2_ (blue circles) obtained in three independent experiments. Each data point represents the mean of an independent experiment (performed in triplicate on different cell passages and on different dates, each consisting of *n* = 3–6 replicates per treatment) and plotted as combined mean ± SD (horizontal lines). + denotes addition of ONO-AE3-208. Statistics: *n*.s., not significant, assessed by a Mann–Whitney test; (**c**) *PTGER4* levels following PGE_2_ stimulation with 10 nM, 100 nM, and 1 µM (pink triangles) for 2 h compared to untreated control (blue circles). Each data point represents the mean of an independent experiment (performed at least in duplicate on different cell passages and on different dates, each consisting of *n* = 3–8 replicates per treatment) and plotted as combined mean ± SD (horizontal lines). SD was not plotted when only two independent experiments were performed. Statistics: $, *p* < 0.05, assessed by a Kruskal–Wallis test with Dunn’s multiple comparisons test; (**d**) *PTGER4* levels following PGE_2_ stimulation for 2 h without concomitant EP2 antagonist (100 nM, pink triangles) compared to controls without PGE_2_ (pink circles), after co-incubation with EP2 antagonist (PF-04418948, 1 µM, orange triangles) compared to controls without PGE_2_ (orange circles) obtained in three independent experiments. Each data point represents the mean of an independent experiment (performed in triplicate on different cell passages and on different dates, each consisting of *n* = 5–6 replicates per treatment) and plotted as combined mean ± SD (horizontal lines). + denotes addition of PF-04418948. Statistics: $, *p* < 0.05; *n*.s., not significant, assessed by a two-tailed Student’s *t*-test.

**Figure 5 cells-09-01256-f005:**
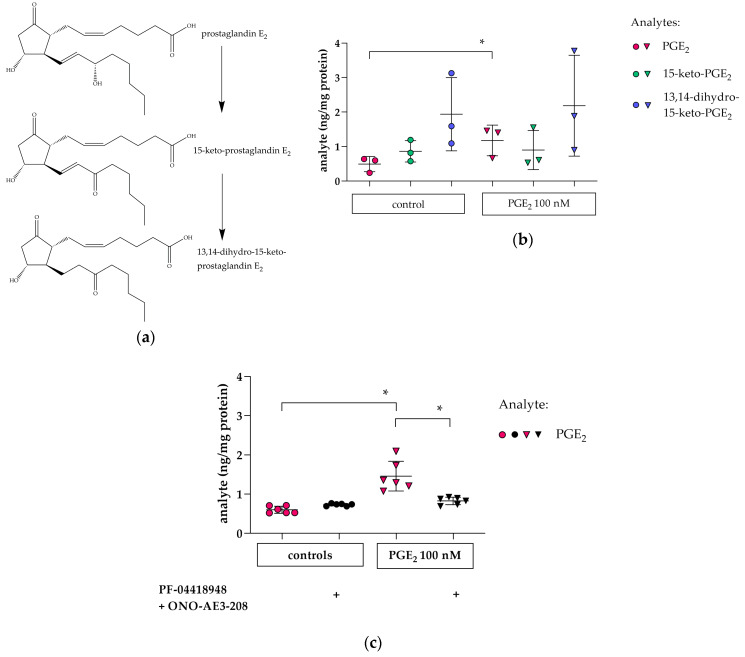
PGE_2_ and its metabolites were measured by LC/ESI-MS/MS in hPC. (**a**) structure of PGE_2_ and its metabolites; (**b**) levels of cellular PGE_2_ (pink), 15-keto-PGE_2_ (green), and 13,14-dihydro-15-keto-PGE_2_ (blue) were measured after PGE_2_ stimulation for 2 h (100 nM, triangles) and in untreated controls (circles). PGE_2_ levels were increased in PGE_2_-stimulated cells (pink triangles) vs. controls (pink circles). Each datapoint represents the mean of an independent experiment (performed in triplicate on different cell passages and on different dates, each consisting of *n* = 3–6 replicates per treatment) and plotted as combined mean ± SD (horizontal lines). Statistics: *, *p* < 0.01, assessed by a two-tailed Student’s *t*-test in each experiment; (**c**) elevated cellular PGE_2_ levels caused by PGE_2_ stimulation (pink triangles) were abrogated by simultaneous co-incubation with combined EP2 and EP4 antagonism (PF-04418948 and ONO-AE3-208, respectively, 1 µM each). + indicates addition of combined EP antagonists. Each datapoint represents a single sample and plotted as mean ± SD (horizontal lines) per treatment group consisting of *n* = 6 replicates obtained in a single experiment. Statistics: *, *p* < 0.01, assessed by a two-tailed Student’s *t*-test.

**Figure 6 cells-09-01256-f006:**
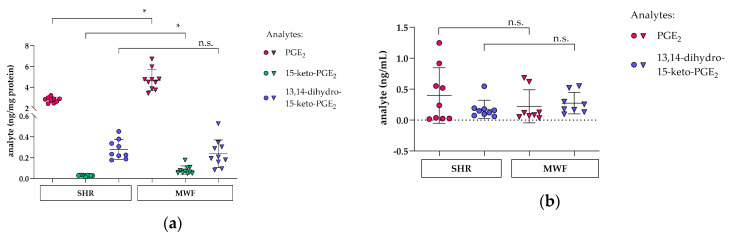
Levels of PGE_2_ (pink), 15-keto-PGE_2_ (green) and 13,14-dihydro-15-keto-PGE_2_ (blue) were measured in glomeruli of MWF (triangles) and SHR (circles) at 8 weeks of age. (**a**) glomerular PGE_2_ and 15-keto-PGE_2_ levels were increased in MWF (pink and green triangles, respectively) compared to SHR (pink and green circles, respectively), whereas glomerular 13,14-dihydro-15-keto-PGE_2_ (blue circles and triangles) did not differ between both strains. Each data point represents a single animal and plotted as mean ± SD (horizontal lines) per rat strain consisting of *n* = 9–10 animals each. Statistics: *, *p* < 0.01; *n*.s., not significant assessed by a two-tailed Student’s *t*-test; (**b**) levels of PGE_2_ (pink) and 13,14-dihydro-15-keto-PGE_2_ (blue) in plasma did not differ between MWF (triangles) and SHR (circles). Each data point represents a single animal and plotted as mean ± SD (horizontal lines) per rat strain consisting of *n* = 8–9 animals, each. Statistics: *n*.s., not significant assessed by the Mann–Whitney test.

**Figure 7 cells-09-01256-f007:**
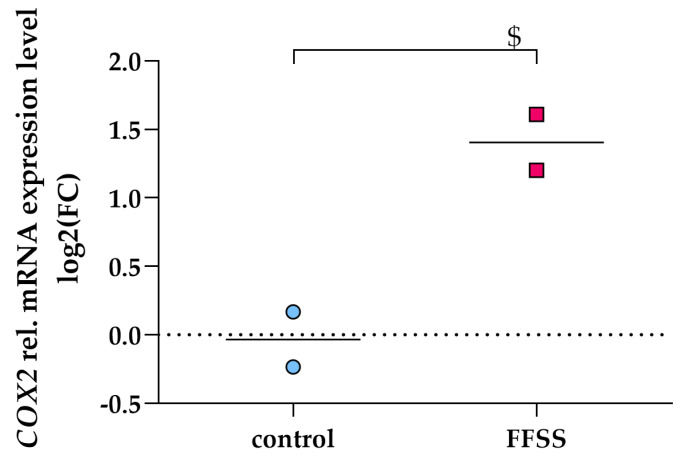
FFSS upregulated *COX2* gene expression in hPC (pink squares). qPCR results are presented as relative mRNA expression level normalized to *GAPDH* and referred to control group (blue circles). *COX2* upregulation was quantified after 2 h of FFSS with 2 dynes/cm^2^. Statistics: $, *p* < 0.05, assessed by a two-tailed Student’s *t*-test. Each datapoint represents the mean of an independent experiment (performed in duplicate on different cell passages and on different dates, each consisting of *n* = 5–6 replicates per treatment) and plotted as combined mean (horizontal lines).

**Figure 8 cells-09-01256-f008:**
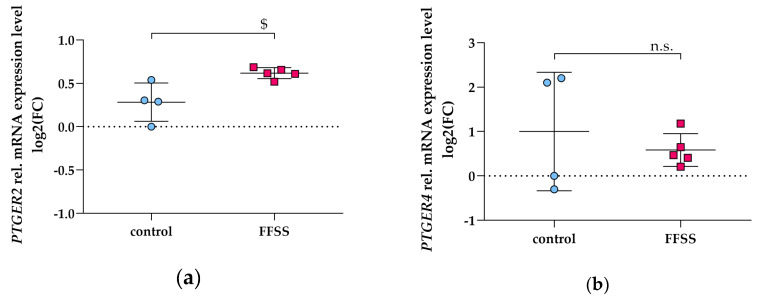
*PTGER2* and *PTGER4* gene expression in hPC subjected to FFSS (pink squares). qPCR results are presented as relative mRNA expression level normalized to *GAPDH* and referred to control group (blue circles). mRNA expression of *PTGER2* (**a**) and *PTGER4* (**b**) was quantified after 2 h of FFSS with 2 dynes/cm^2^. Statistics: $, *p* < 0.05, assessed by a two-tailed Student’s *t*-test. Each datapoint represents a single sample and plotted as mean ± SD (horizontal lines) per treatment group consisting of *n* = 4–5 replicates obtained in a single experiment.

**Table 1 cells-09-01256-t001:** Primer sequences for human (h) genes of interest.

Gene	Forward Primer (5′-3′)	Reverse Primer (5′-3′)
*hGAPDH*	gagtcaacggatttggtcgt	gatctcgctcctggaagatg
*hCOX2*	tgatgattgcccgactcccttg	tgaaagctggccctcgcttatg
*hPTGER1*	ttcggcctccaccttctttg	cgcagtaggatgtacacccaag
*hPTGER2*	gacggaccacctcattctcc	tccgacaacagaggactgaac
*hPTGER3*	tctccgctcctgataatgatg	atctttccaaatggtcgctc
*hPTGER4*	ttactcattgccacctccct	agtcaaaggacatcttctgcca

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
