# Peer review of "Concerted EP2 and EP4 Receptor Signaling Stimulates Autocrine Prostaglandin E2 Activation in Human Podocytes"

_cells, 2020, doi:10.3390/cells9051256_

Round 1

Reviewer 1 Report

Mangelsen E et al in their manuscript “Concerted EP2 and EP4 receptor signaling stimulates autocrine Prostaglandin E2 activation in human podocytes” use human podocytes and the Munich Wistar Frömter rat, a model for CKD to show the following that add to the current literature:

(a) Human podocytes like murine podocytes express EP1, EP2 and EP4.

(b) Both G-coupled EP2 and EP4 receptors stimulate adenylate cyclase activity resulting in elevated cAMP (Figure 2) and in PGE2 mediated amplification of Cox2 (Figure 3) in vitro experiments.

(c) The response of human podocytes to FFSS is like the murine podocytes for COX2 (Figure 5).

(d) They demonstrate the elevated PGE2 and 15-keto-PGE2 in the glomerulus of MWF rats using LC/ESI-MS/MS as a direct evidence of increased PGE2 within the glomerulus (Figure 7).

The authors’ work adds to the new evolving concept of FFSS in hyperfiltration-mediated injury. I have a few minor comments for their consideration:

(i)              The current work is about EP2 and EP4. I would like to see the EP1-4 expression mRNA expression data from their rat experiment (they only report COX2). The biological samples exist to run these experiments and are needed to complete their story.

(ii)             The authors refer to the murine experiments where FFSS has been shown to change mRNA expression of EP2, EP4 and COX2. The current work is about EP2 and EP4. I would like to see the EP2 and EP4 mRNA expression data from their FFSS and PGE2 experiments. The biological samples exist to run these experiments and are needed to complete their story.

(iii)           Figure 4(c) does not show changes in PGE2 for EP2 antagonist and EP4 antagonist alone in line with their cAMP and COX2 studies. They need to be part of the Figure 4(c), especially when these antagonists independently decrease the COX2 mRNA expression. These experiments and are needed to complete their story.

(iv)           It would be necessary to show a dose response for selection of 1 µM for EP2 and 1 µM for EP4 antagonists. It can be provided in the Supplement Section.

(v)             IBMX is added to the cells to inhibit phosphodiesterase enzyme in cAMP experiments. Although it may not be critical in a 20-min experiment, but this needs to be added to the discussion as a limitation.

(vi)           In their FFSS experiments they state “culture slips” on line 217. Do they mean “culture glass slides”?

(vii)          In Supplementary File Figure 2, in contrast to literature where EP4 is constitutively expressed in large amounts in podocytes and EP2 is a low expression protein, the expression of EP2 is more intense than EP4. Can you please discuss this observation in human podocytes?

(viii)        The results could be further strengthened by showing increased cAMP and COX-2 with EP2 and EP4 agonists without PGE2 to validate the role of EP2 and EP4 receptors.

Reviewer 2 Report

This paper by Eva Mangelsen and colleagues investigate the role of COX2/PGE2 pathway in podocyte injury in a rat model of spontaneous albuminuria and immortalized human podocytes. This is a field with major need for improvement as CKD is becoming a great challenge for public heath and kidney diseases are one of the major chronic complications associated with increased morbidity and mortality. Overall, the paper is well written, text is precise, concise and should constitute interesting reading to wide group of investigators. Unfortunately, multiple parts of the manuscript concern the reviewer including: the lack of novelty, the lack of convincing evidence of podocyte injury, a questionable data presentation, and very limited characterization of utilized cell line.

Here are some suggestions and comments that the authors should consider:

  • The novelty of the presented study is somehow questionable as there are already multiple reports providing evidence for COX2/PGE2 pathway in podocyte physiology and pathophysiology. Additionally, it seems that there is a great deal of species-dependent differences.
  • Introduction section: This part of the manuscript is very dense and heavy on raw facts, which makes it very difficult to follow. It would be very helpful for the reader if the authors could simplify that part and rewrite it focusing only on the essential information that is needed to follow the story. The current shape does not allow to clearly identify the reason why the study was conducted.
  • hPC cell line has been used in the study, however the characteristics of the cells are somehow missing. It is well know that immortalized cell line of different cell types may not express characteristics of specific cell. The structure of podocytes presented on figure S1 lacks typical “stretchy” foot processes. It would be essential if authors provided evidence for other makers of podocytes (i.e. podocin, nephrin, ZO-1).
  • Ketamine has been shown to have inhibitory effects on multiple types of receptor and responses (i.e. purinoceptors, IL-6, NF-kB). Is that something that was taken under consideration for that study?
  • The authors associate PGE2 pathway with podocyte damage and conclude that “this contributes to disturbances of glomerular filtration barrier and the development of albuminuria” No evidence of increased GFR, podocyte damage, and resultant albuminuria are provided, therefore it seems to be a great overstatement and not accurate interpretation of the results.
  • It is somehow striking that the author did not attempt to isolate podocytes from MWF rats and measure the components of COX2/PGE2 axis. Glomeruli contain multiple different types of cells. What is known about the expression/production of COX2 and PGE2 by other glomerular cells (i.e. mesangial cells, endothelial cells). The authors summarize that: “elevated PGE2 and 15-keto-PGE2 levels in glomeruli of MWF support the hypothesis that glomerular accumulation of PGE2 is associated with albuminuria due to podocyte damage…’’. Podocyte-specific injury and consequent albuminuria has not been shown.
  • One of the major concerns of the reviewer is data presentation. Data analysis is one of the most crucial parts of the studies since authors are trying to determine the role of COX2/PGE2 in podocyte pathology. Some of the figures (Fig.3a, 5) present n=2 with SD (for n=2 SD should not be presented). The authors explain in the figure legend that each data point represents the mean of independent experiment, performed in duplicates. Regarding Fig 3a, are those cells collected from the same passage and just plated on separate plates? If so these are just replicates not independent experiments. Regarding Fig 5, what exactly is n=5-6? Is it a number of glomeruli or is it a number of rats that the glomeruli were obtained from? If so it would be essential for the data transparency purposes to present all individual points (the same as on Fig 6).
  • The authors make a lot of associations to hyperfiltration and its potential contribution to podocyte damage. What is known about GFR values in 8 weeks old MWF rats?

Minor:

  • A lot of abbreviations that are not explained in the text.

Round 2

Reviewer 1 Report

They have addressed all the points raised by me. I would recommend accepting the article.  

Reviewer 2 Report

The authors did a good job with addressing all the the raised issues. Thank you.